# The Early Prediction of Common Disorders in Dairy Cows Monitored by Automatic Systems with Machine Learning Algorithms

**DOI:** 10.3390/ani12101251

**Published:** 2022-05-12

**Authors:** Xiaojing Zhou, Chuang Xu, Hao Wang, Wei Xu, Zixuan Zhao, Mengxing Chen, Bin Jia, Baoyin Huang

**Affiliations:** 1Department of Information and Computing Science, Heilongjiang Bayi Agricultural University, No. 5 Xinyang Road, Daqing 163319, China; zhouxiaojing7924@126.com; 2Heilongjiang Provincial Key Laboratory of Prevention and Control of Bovine Diseases, College of Animal Science and Veterinary Medicine, Heilongjiang Bayi Agricultural University, No. 5 Xinyang Road, Daqing 163319, China; zzx032607@163.com (Z.Z.); chenmengxing0614@163.com (M.C.); 3Animal Husbandry and Veterinary Branch of Heilongjiang Academy of Agricultural Science, Qiqihaer 161005, China; tlwanghao777@126.com (H.W.); wenxi311@126.com (B.J.); 18249556543@163.com (B.H.); 4Department of Biosystems, Division of Animal and Human Health Engineering, KU Leuven, 3000 Leuven, Belgium; weixu@kuleuven.be

**Keywords:** disorders, machine learning, prediction, rumination, milk yield, electrical conductivity of milk

## Abstract

**Simple Summary:**

Identifying cows with a higher risk of health disorders such as clinical mastitis, subclinical ketosis, lameness, and metritis could be advantageous for farms to prevent and ameliorate the negative effects of these disorders in a timely manner. In this study, we adopt eight machine learning algorithms using an R software for analyzing a dataset of 14-dimensions of dairy cows with health disorders across the whole lactation period in intensive Chinese dairy farms, applying automatic monitoring systems and milking systems. The variables analyzed by the machine learning algorithms include milk yield, physical activity, changes in rumination time, and the electrical conductivity of milk. Six parameters were presented to evaluate the performance metrics of the models, with the Rpart algorithm outperforming others and indicating a strong generalization ability of this algorithm. A total of 10 variables of greater importance in three models of Rpart, eXtreme Gradient, and Adaboost demonstrated the consistency of those variables as predictors for disorders of dairy cows monitored by automatic systems. The results obtained in this study highlighted the importance of using big data on the farm to develop predictive and prescriptive decision support tools to boost the development of precision livestock farming.

**Abstract:**

We use multidimensional data from automated monitoring systems and milking systems to predict disorders of dairy cows by employing eight machine learning algorithms. The data included the season, days in milking, parity, age at the time of disorders, milk yield (kg/day), activity (unitless), six variables related to rumination time, and two variables related to the electrical conductivity of milk. We analyze 131 sick cows and 149 healthy cows with identical lactation days and parity; all data are collected on the same day, which corresponds to the diagnosis day for disordered cows. For disordered cows, each variable, except the ratio of rumination time from daytime to nighttime, displays a decreasing/increasing trend from d-7 or d-3 to d0 and/or d-1, with the d0, d-1, or d-2 values reaching the minimum or maximum. The test data sensitivity for three algorithms exceeded 80%, and the accuracies of the eight algorithms ranged from 65.08% to 84.21%. The area under the curve (AUC) of the three algorithms was >80%. Overall, Rpart best predicts the disorders with an accuracy, precision, and AUC of 81.58%, 92.86%, and 0.908, respectively. The machine learning algorithms may be an appropriate and powerful decision support and monitoring tool to detect herds with common health disorders.

## 1. Introduction

Automatic monitoring encompasses substantial data on rumination time, physical activity, and feeding behavior and has been applied across many intensive farm systems worldwide. These data are often coupled with milk yield-related variables monitored by automated milking systems. Metritis, mastitis, metabolic, and digestive and hoof disorders are common in intensive dairy farms, and the majority of cases may considerably affect the welfare of dairy cows [1] and reduce farm profitability. They can cause a decline in the milk yield [2,3], increased rate of culling and death [4,5,6], high treatment costs [7], and impaired reproductive performance [8,9,10].

The incidence of some health problems in early lactation has been linked to an altered curve with a lower peak of milk yield (PMY), delayed PMY, and reduced milk production during the entire lactation. Moreover, this negative effect may persist in the subsequent lactations [11,12]. Owing to the poor awareness of the early prevention of health disorders and detection techniques among farmers, most subclinical diseases remain undiagnosed until the onset of evident clinical symptoms, thus making treatment difficult and expensive while prolonging the negative effects on the health and performance of dairy cows. Thus, data preceding diagnoses that can predict the risk or detect disease more efficiently than clinical signs would be inherently more useful for earlier detection and intervention. 

Machine learning is a subfield of AI (artificial intelligence) and adopts statistical techniques to detect or predict cow performance or disease events using large datasets and can deal with complicated correlations caused by an ever-increasing number of variables [13]. Researchers have used machine learning algorithms to detect or predict several health disorders, such as clinical mastitis using random forest, naïve Bayes, and eXtreme Gradient boosting [14], neural networks [15,16], decision-tree induction [17], and logistic generalized linear mixed models [18]. For claw lesions and lameness, researchers have applied random forest [19,20], K nearest neighbors [21], decision tree [22], and naïve Bayes [19]. Moreover, metritis cure [23] and metabolic status [24] of dairy cows in early lactation have been simultaneously implemented by seven or eight machine learning algorithms. Morteza et al. explored different metabotypes of dairy cows in the transition period using the decision tree, random forest, and naïve Bayes [25]. In addition, the K nearest neighbor, decision tree, and multilayer perceptron were used by Warner et al. [22] to discriminate abnormal behavior in dairy cows with subacute ruminal acidosis.

In northeast China, with the intensive development of dairy farms, fewer workers and increased workload have created great difficulties in early monitoring, detection, and diagnosis of dairy bovine diseases; particularly, an early warning using the data generated by automatic systems is lacking. The research problem is the ability to accurately and timely predict and/or detect disorders in dairy cows with live data on intensive farms. In machine learning, it is a common practice to evaluate several algorithms on integrated data (e.g., management, health, milking) since the performance of each algorithm may depend on features, sample size, structure, and other characteristics of the data set [24]. In this study, we aimed to construct prediction models for naturally occurring health disorders in dairy cows monitored by automated systems (neck collar, milking system) based on eight machine learning algorithms. We intended to evaluate each prediction model with datasets collected from two commercial farms in northeast China.

## 2. Materials and Methods

This research was a part of a large study aimed at the continuous monitoring, prediction, and early detection of health disorders in cows in commercial herds in the northeast of China using automated monitoring systems and milking systems. Holstein cows with DIM (days in milk) in their entire lactation were enrolled for the experiment.

### 2.1. Animals, Housing and Feeding

We collected the original data from two commercial farms with a straight-line distance of 104.9 km in northeast China from January 2020 to October 2021. The farms were located at a longitude of 121.11 E and 135.05 E and a latitude of 43.26 N and 53.33 N, with a cold temperate zone and temperate continental monsoon climate. The annual average temperature ranged from −5 °C to 5 °C. These farms are the practice base of the Heilongjiang Bayi Agricultural University. Each evaluated group was housed in uniform pens with identical characteristics as follows: (i) enclosed barns on concrete solid floors and sawdust with space for 50–136 cows by providing similar cubicles; (ii) feeding and water area in each pen; (iii) feed bunk and access to an exercise yard for 1 h/day, except in winter. The barns were ventilated naturally and with sprayers in summer at an interval of 20 m. In winter, they were equipped with pipes filled with hot water to alleviate the cold. The farm staff clean the barn when necessary, protect against mosquitoes and flies in summer, and broadcast different types of music to the cows at the specified time, e.g., eating, exercising in the yard, milking in the milking parlor, etc.

The farms applied a total mixed ration (TMR) to feed the cows twice daily (0500 h and 1300 h) with the feed pushed whenever necessary and freely available freshwater. They were milked thrice daily (0300 h, 1100 h, and 1900 h) via a milking system (FreeFlow, SCR Engineers Ltd., Netanya, Israel). Before calving, the cows had ad libitum access to prepartum and postpartum TMR composed of a diet forage-to-concentrate ratio between 78:22 and 60:40, on a dry matter basis. Overall, the management modes and feeding patterns were similar among the involved herds.

We collected sensor monitoring information about the behavioral patterns of all dairy cows, including physical activity coupled with rumination. Moreover, we recorded their performance parameters by the milking system. We selected 900 cows with 308,100 days of observations via the neck collar and milking system between farms 1 and 2 to construct the predicting models. From January 2020 to October 2021, we continuously monitored the records of 298 healthy (without any disease during the experiment) and 244 sick cows with 286 sickness events, and data were collected once a week. We excluded the cows with missing values of the variables monitored by the automatic systems and those transferred to another group more than thrice. Eventually, the final data set included 149 healthy cows and 131 with diseases and 14 variables, including parity, season during the disorder, days in milk (DIM), age at the time of disorder, milk yield, activity, and six variables related to the rumination time, and two variables related to the electrical conductivity of milk. 

### 2.2. Data Collection and Study Design

Data of the physical activity and rumination time recorded with the HR-Tag monitoring system (SCR Engineers Ltd., Netanya, Israel) were averaged and stored at 2-h intervals up to 24 h. Activity was determined by a 3-axis accelerometer that was recorded as a unitless measure of upward vertical head and neck movements, such as walking and mounting, while excluding downward vertical and horizontal movements, such as feeding [3]. The performance data was generated and automatically saved during each milking in the herd management software (DataFlow, SCR Engineers Ltd.). Moreover, we calculated the daily milk production as the sum of all milk collected per cow per day, similar to the activity and rumination time variables. Table 1 summarizes the variables used for the statistical analysis.

During the entire lactation, primiparous and multiparous cows were fed a similar TMR. They were regularly checked for signs of disease or injury by the farm staff as well as for reproductive events and survival. Warning events predicted by the automated monitoring system were determined and health assessments were routinely performed during the transition period, e.g., hoof trimming and body condition score assessment. The herd veterinarians and the farm technicians recorded all cases of health problems, diagnoses, and treatments according to standard operating procedures of the farm.

During data editing, the following criteria were used to remove records from the following final data set: parameters monitored by the milking systems and automatic monitoring systems without data; cows with missing data 14 days before the diagnosis in a group; cows that were moved between herds more than twice within a lactation period. We analyzed the milk yield, physical activity, rumination time, and the electrical conductivity of milk of healthy and sick cows. We hypothesized that the aforementioned variables would begin to deviate from normal from 3 to 7 days or more before the diagnosis; thus, they would facilitate detection. Data of healthy cows were collected with the same parity and similar DIM as those with disorders. The date denoted as d0, i.e., the day from which the milk yield, activity, and rumination time of 8 days were collected, corresponded to the diagnosis day of those with disorders.

### 2.3. Defining Health Disorders

We identified metritis cases by a foul-smelling vulvar discharge and a rectal temperature ≥39.5 °C in cows at 3 days and 21 days following calving or during the entire lactation. Clinical signs of mastitis were regularly examined by observing the udder and milk (i.e., hard quarters, heat or swelling, clots in milk, flakes, or lumps, or clear/yellow milk) following calving until day 28 and were subsequently determined every 3 days throughout the lactation. Veterinarians diagnosed cases of hoof disorders, including digital dermatitis, interdigital dermatitis, sole ulcers, and abscesses upon detecting mechanical or infectious lesions or during routine hoof trimmings. They were treated with corrective hoof trims. Poor appetite, scant manure, and ruminal and intestinal stasis, including ruminal indigestion, forestomach retardation, and ruminal flatulence, indicate digestive disorders. Our research team defined the health-monitoring program before the start of this study, and the farm staff (for each farm, 1 manager, 3 technicians, and 1 veterinarian with more than 15 years of experience monitoring cow health) were responsible for conducting the daily health monitoring of the dairy cows. The time from detection to diagnosis should not exceed 6 h, and the information (which comprises the cow identification number, the date of diagnosis, the type of disorder, and the staff who detected and diagnosed the disorders) of the animals were inputted into the management system software within 5 min after diagnosis. We did not consider a cow suffering from two disorders at the same time or sick more than once during one lactation.

### 2.4. Statistical Analyses and Machine Learning Algorithms

The cows were initially grouped into the following two categories: disordered and healthy, with the day of diagnosis and treatment of each disorder considered as d0. We collected data for the following variables from 7 days (d-7) to 3 days (d-3) before diagnosis to d0: daily milk yield (kg/day), daily activity (unitless), daily rumination time (min/day), rumination time at daytime (min/day), rumination time at nighttime (min/day), the ratio of rumination at daytime to nighttime (unitless), rumination deviation per 2 h (difference between recorded and expected rumination time, for which recorded every 2 h and transferred to the automated health-monitoring system software), the sum of the absolute values of weighted rumination time variation (unitless, described the stability of rumination time and was calculated by private algorithms developed by the company providing the sensors), the daily percentage of the electrical conductivity of milk change (unitless), and the peak electrical conductivity of milk (mS/cm). The aforementioned variables were statistically analyzed unless otherwise stated. Cows with disorders were classified into subgroups according to their lactation stage (DIM1-100, 101-200, and 201-dry off denoted as lactation stage 1, lactation stage 2, and lactation stage 3, respectively), season (cold season from January to April, November and December, and hot season from May to October, according to the climate of the selected farms), and parity (one, two, and ≥three).

Before data categorization into the training set and testing set, we performed descriptive statistics to characterize the measures of location and variability by means of frequency distribution tables and histograms. Thereafter, we performed the x^2^ tests and *t*-test for categorical outcomes and continuous variables, respectively. *p*-values < 0.05 denote statistical significance (trends declared at 0.05 < *p* ≤ 0.10).

We performed machine learning algorithms using the R software version 4.1.2 (R Core Team, 2021, https://www.r-project.org/, accessed on 3 April 2021) as well as data processing. Moreover, we adopted the “t.test ()” function for Pearson analysis. For logistic regression, “glm” was adopted with the parameter “family” selected as the “binomial”. For the decision tree, we used “rpart,” and for “parms”, considering split principle, “gini” was chosen, and the parameter of complexity “cp” was set as 0.001 to receive relatively stronger punish power and simpler “tree” following repeated parameter adjustment. For eXtreme Gradient, we adopted the “xgboost” package and set “fullRank” in the model as “TURE” to exclude complete collinearity and “adabag” for adaboost. The “mfinal” parameter was set as 1000, considering the data sample, “e1071” for support vector machine with “sigmoid” selected as the “kernel,” and gama set as 0.1 following multiple rounds of debugging. Moreover, we used the “randomforest,” “klaR,” and “kknn” functions for random forest, Naive Bayes, and k-nearest neighbor algorithm (KNN), respectively.

To ensure the repeatability of our results, “set seed ()” was set for the above-mentioned algorithms. For each model, we randomly divided the data according to the dependent variable “Species” (binary variable, “0” represented “healthy cows” vs. “1” “disordered cows”) using the “createDataPartition” function. We selected a data subset consisting of 75% of the observations as the training data to construct the predicting models. By contrast, the remaining 25% was used as the test data to assess the performance of the models, which were trained using 10-fold cross-validation. For the “confusionMatrix” function, “positive” and “mode” were set as “disordered” and “everything” to output the maximum metrics for evaluating model performance.

We assessed the performance of each machine learning algorithm by their sensitivity, specificity, accuracy, precision, F1-score, and area under the receiver operating characteristic (ROC) curve (AUC) value. The AUC (95% confidence interval) was defined as follows:(1)$$\mathrm{Sensitivity}\;=\frac{\mathrm{TP}}{\mathrm{TP}+\mathrm{FN}},\;\mathrm{Specificity}\;=\frac{\mathrm{TN}}{\mathrm{TN}+\mathrm{FP}},\;\mathrm{Accuracy}\;=\frac{\mathrm{TP}+\mathrm{TN}}{\mathrm{TP}+\mathrm{TN}+\mathrm{FP}+\mathrm{FN}},$$
(2)$$\mathrm{Precision}=\frac{\mathrm{TP}}{\mathrm{TP}+\mathrm{FP}},\;F1-\mathrm{score}=\frac{2*\mathrm{Precision}*\mathrm{Sensitivity}}{\mathrm{Precision}+\mathrm{Sensitivity}}$$
where, true positives (the number of cows with actual health disorder, predicted as with disorder), false negatives (the number of cows with actual health disorder, predicted as healthy), true negatives (the number of healthy cows, predicted as healthy), and false positives (the number of healthy cows, predicted as with health disorder) were denoted as TP, FN, TN, and FP, respectively.

Table 2 summarizes the values of the eight algorithms.

## 3. Results

We analyzed 131 sick cows (10, 29, 59, and 33 cows with a digestive disorder, lameness, mastitis, and metritis, respectively), with an average DIM of 104.45 ± 95.147, parity of 3.01 ± 1.571, and age at suffering disorder of 48.81 ± 13.986. We collected data for 149 cows in the control group with almost similar lactation days (108.35 ± 101.290), parity (3.02 ± 1.631), and age (45.29 ± 12.374).

### 3.1. Variation Analysis of Each Variable for Dairy Cows with Health Disorders

For cows with health disorders, each variable, except the ratio of the rumination time at daytime to nighttime, displayed a decreasing/increasing trend from 7 days or 3 days before diagnosis (d-7 or d-3) to the diagnosis day (d0). Moreover, the value of d0 or d-1 or d-2 reached the minimum or maximum. Milk yield displayed a decreasing trend from d-7 to d0. The yield was 2.07 kg and 5.85 kg lower on d-1 (*p* < 0.001) and d0 (*p* < 0.001), respectively, than the average milk yield from d-7 to d-2. The total daily rumination time on d-1 and d0 was 39.42 min (*p* = 0.001) and 45.53 min (*p* = 0.001) less, respectively, than the average rumination time from d-7 to d-2. Moreover, it displayed a decreasing trend from d-7 to d0. The rumination time at nighttime reached the minimum on d-1 (282.62 min) and was 30.65 min less than the average time from d-7 to d-2 (*p* < 0.001). The ratio of the rumination time from daytime to nighttime was 12.48 (*p* < 0.001) and 9.51 (*p* < 0.001) higher on d0 and d-1, respectively, than the average ratio from d-7 to d-2. The deviation of the rumination time per 2 h was higher on d0 than that from d-7 to d-1, and this variable displayed an increasing trend from d-7 to d0. We identified a rapid increase in the sum of absolute values of the weighted variation of the rumination time from d-3 to d0 for cows with disorders. The peak electrical conductivity of milk displayed an increasing trend from d-3 to d0 (*p* < 0.001) and reached its highest on d0 (5.87 ± 0.80). Moreover, the daily percentage of the electrical conductivity of milk changed had the highest value of 108.94 ± 12.00 on d0. While there was no significant difference among the cows classified according to the DIM, season, parity, and age of those with disorders.

### 3.2. Difference Analysis of Each Variable for Those with Disorders and Healthy Ones

Except for the ratio of the rumination time at daytime to nighttime, other variables were significantly different between the disordered and healthy groups on d-1 and/or d0. The average daily milk yield of the disordered group was 1.76 kg (*p* < 0.001) and 5.41 kg (*p* < 0.001) less than that of the healthy group on d-1 and d0, respectively. The daily rumination time for the disordered group was significantly less than that for the healthy group from d-3 to d0 (*p* < 0.001). Moreover, the rumination time for the disordered group was 62.64 min and 57.02 min less than that for the healthy group on d0 and d-1, respectively. The rumination time at daytime and nighttime displayed a similar difference between the groups. The ratio of the rumination time from daytime to nighttime was higher for the disordered group than for the healthy group from d-7 to d0. Moreover, the groups displayed significantly different values on d-7 and d-3 (*p* < 0.001). Particularly, the sum of absolute values of weighted rumination time variations was different between the groups from d-7 to d0 (*p* < 0.001). The largest difference was observed on d0 (68.15 ± 55.02), followed by d-1 (60.84 ± 35.37). The average value of the peak electrical conductivity of milk in healthy cows was 5.36 ± 0.10 from d-3 to d0, which was significantly lower than that in those with the disorder (*p* < 0.001). The average difference was 0.39 ± 0.05, with the largest difference of 0.50 ± 0.07 on d0. We observed a similar difference in the daily percentage of the electrical conductivity of milk change, thus confirming the electrical conductivity of milk as an indicator for the early detection of health disorders in dairy cows. There was no evident increasing/decreasing trend in the total daily activity for the disordered group, whereas their average was significantly lower than that for healthy ones from d-7 to d-1. We obtained opposite data on d0 supposedly owing to the involvement of the estrus. Overall, the aforementioned variables could be used for predictive models for the early detection of health disorders in dairy cattle. Figure 1 depicts box plots with the error bar and significance of each variable.

### 3.3. Performance of Machine Learning Algorithms

We evaluated the performance of each machine learning algorithm according to its sensitivity, specificity, accuracy, F1−score, and AUC value (Table 2). The specificity of the SVM, Rpart, random forest, and eXtreme Gradient models exceeded 80%. By contrast, the sensitivity (recall) of these models did not meet the expected value. The sensitivity of only the random forest reached 83.33%. The accuracy of the three models was >80%, namely, SVM, Rpart, and random forest. The precision of Rpart reached 92.86%, and no algorithm exhibited overfitting.

The overall AUC values for the aforementioned three models were >0.80, with Rpart displaying the highest AUC (0.908). The performance metric of Rpart classification was better than that of the remaining models, which indicated its powerful and credible generalization ability. Figure 2, Figure 3 and Figure 4 depict its rank of variable importance and that of eXtreme Gradient and Adaboost. Figure 5 represents the ROC curves of the train data and test data of Rpart, and the threshold was obtained by the Youden index (sensitivity + specificity-1).

The ten variables, namely, aweid1, aweid3, nr1, rd1, rd2, rdn1, md1, ecv2, aweid2, and ecp2, represent the sum of the absolute values of weighted rumination variation on d-1, the sum of the absolute value of weighted rumination variation on d-3, rumination time at nighttime on d-1, the daily rumination time on d-1, the daily rumination time on d-2, the ratio of the rumination time at daytime to nighttime on d-1, the daily milk yield on d-1, the daily percentage of the variation of electrical conductivity of milk on d-2, the absolute value of weighted rumination variation on d-2, and the peak electrical conductivity of milk on d-2, respectively. The *x*-axis represents the score of the importance of the features evaluated by the measure “Gini” (Figure 2).

The ten variables, namely, rd1, ecv1, ecp2, aweid1, nr4, ecp1, md1, rdn2, rd2h3, and ecv3, represent the daily rumination time on d-1, the daily percentage of variation of the electrical conductivity of milk on d-1, peak electrical conductivity on d-2, the sum of the absolute value of weighted rumination variation on d-1, rumination time at nighttime on d-4, the peak electrical conductivity on d-1, the daily milk yield on d-1, the absolute value of rumination deviation every 2 h on d-3, and the daily percentage of variation of electrical conductivity of milk on d-3, respectively. The *x*-axis represents the score of the importance of features evaluated by the measure “Gini” (Figure 3).

The ten variables, namely, aweid1, aweid3, aweid2, ecv1, rd2h3, nr1, ecv3, rd2, rd2h1, and nr2, represent the sum of the absolute value of weighted rumination variation on d-1, the sum of the absolute value of weighted rumination variation on d-3, the sum of the absolute value of weighted rumination variation on d-2, the daily percentage of change of the electrical conductivity of milk on d-3, the daily percentage of variation of the electrical conductivity of milk on d-1, the absolute value of rumination deviation every 2 h on d-3, daily rumination time on d-2, the absolute value of rumination deviation every 2 h on d-1, and rumination time at nighttime on d-2, respectively. The *x*-axis represents the score of the importance of features evaluated by the measure “Gini” (Figure 4).

The black line depicts the ROC curve for train data, with the area under the ROC curve, whereas the red line depicts the curve for test data.

The logistic regression model was based on the variables comprising daily milk yield (md3, md2, and md1), daily rumination time (rd3, rd2, and rd1), the ratio of daytime to nighttime (rdn3, rdn2, and rdn1), the sum of absolute values of weighted rumination variations (aweid3, aweid2, and aweid1), peak electrical conductivity (ecp3, ecp2, and ecp1), and six significant variables with the coefficient and standard error of ecp1 (1.0694 ± 0.3068, *p* < 0.001), md3 (0.0811 ± 0.0479, *p* < 0.1), rd2 (−0.0144 ± 0.0034, *p* < 0.05), rdn1 (1.8941 ± 0.9172, *p* < 0.05), aweid3 (0.0117 ± 0.0058, *p* < 0.05), and aweid1 (0.01168 ± 0.0048, *p* < 0.05). Moreover, the ecp1 variable (peak electrical conductivity of milk on d-1) increased by one unit, and the risk of disorders was likely to increase by 2.91 ± 1.36 times (*p* < 0.001) compared with that in cows without an increasing peak electrical conductivity of milk.

## 4. Discussion

Identifying cows at a higher risk of health disorders such as clinical mastitis, subclinical ketosis, lameness, and metritis could be advantageous for farms. This is because timely actions can prevent and ameliorate the negative effects of these disorders. This, in turn, will facilitate health management for individual dairy cows and the entire herd in a precision farming system [26].

Several researchers have explored machine learning algorithms to detect or predict the onset of several health disorders. However, each study has focused on only one or two disorders, and the majority of them involved blood samples. Support vector machines have been employed to detect clinical mastitis in Holstein cows milked with automated milking systems. An accurate classification rate reached >90% [27], which outperformed the one obtained in the current study. In their study, Fadul-Pacheco et al. reported a random forest algorithm with the best performance for predicting clinical mastitis, with a sensitivity and specificity of 85% and 62%, compared with 68.42% and 98.74%, respectively, in our study [14].

The somatic cell counts (SCC) and LogSCC have been considered predictors of clinical mastitis [28]. The participating farms measured the SCC of the entire herd only once a month, and they likely did not facilitate the detection or prediction of disorders. Thus, we did not consider the aforementioned variable. Using the five biochemical indicators of insulin, free fatty acids, plasma beta-hydroxybutyrate, plasma glucose, and insulin-like growth factor 1, Xu et al., predicted the metabolic status of dairy cows from lactation week 1 to 7 postpartum via eight machine learning algorithms with random forest and support vector machine outperforming other algorithms [24]. The invasive nature of blood sampling limits its applicability in large commercial farms and is generally impractical for the detection of several disorders in the entire herd. In the current study, we did not consider any biochemical indicators in the herd monitored by the automatic system or year-round data.

Steensels et al., proposed a decision tree algorithm to identify 35 post-calving cows with disorders (ketosis and/or metritis) with sensor data, including the performance data, rumination time, cow activity, and body weight [29]. In the current study, 33 cows had metritis during the entire lactation and not just the 28 days following calving, despite the higher proportion of detection in early postpartum cows. The first four variables identified by Steensels et al., were similar to those in this study. Moreover, we introduced three and two variables related to the rumination time and the electrical conductivity of milk, which resulted in Rpart outperforming other algorithms.

Morteza et al., clustered different metabotypes of cows in the transition period by applying four machine learning algorithms after combining their body condition score, backfat thickness, and 11 blood samples collected weekly to assess the serum concentrations of metabolites, with the accuracy exceeding 70% for all algorithms [25].

To evaluate metritis treatment, de Oliveira et al., proposed seven machine learning models with the best performance by a random forest classifier [23]. The milk difference, temperature, treatment, week postpartum, and vulvovaginal laceration were included in the model, which were unavailable in the current experiment. Nevertheless, the random forest classifier was the better model, with a generalization ability and sensitivity, specificity, accuracy, and precision of 83.33%, 84.62%, 84.21%, and 71.43%, respectively. Warner et al. [22] adopted the KNN, decision tree, and multilayer perceptron to detect abnormal behavior in dairy cows (28 cows) with subacute ruminal acidosis. The KNN performed the best among the three algorithms; in contrast, it displayed the lowest metrics in our study, probably because of the involved sample and the variables selected in the prediction model.

Shahinfar et al., used naïve Bayes, random forest, and multilayer perceptron to predict lameness in dairy cows by analyzing four types of production and phenotypic data with 20 subcategories of indicators [19]. They compared three algorithms with linear regression, and random forest displayed the best performance. In contrast, Rpart overperformed random forest and naïve Bayes in our study. Our results of logistic regression models were consistent with those reported by van Hertem et al., for detecting lameness based on the multivariate continuous sensing of neck activity and rumination, coupled with the yield of Holstein milking cows on one Israeli farm [30]. Important variables in both studies included the daily milk yield, the ratio of nighttime to daytime neck activity, and ruminating time during the nighttime from d-7 to d0. In contrast, we specified four types of disorders, not a single disorder, which may have affected the performance and generalization ability of the adopted machine learning algorithms. Similar results were obtained by Cavero et al. [16]; however, we did not achieve the expected results using an artificial neural network for the detection of health disorders with the analyzed variables as input data.

The early detection of cows at risk of disorders could allow for timely intervention, thus decreasing their negative effects. Health disorders in dairy cattle can be identified and predicted using machine learning algorithms by integrating and analyzing data related to the milk yield, physical activity, and changes in rumination time monitored by several sensors. However, there is a severe paucity of data on the use of behavior, rumination time, and productivity for the early prediction or detection of disorders in Chinese intensive dairy farms. These farms have widely adopted automated systems, which underscores the potential benefit of using the data as a predictive tool.

Fine-tuning the prediction model by adjusting the contribution of the variables could improve the predictive performance and maximize the use of data collected on/off the farm to generate farm-based algorithms based on the farmer’s needs, management mode, and conditions. This study pioneered the aforementioned idea by exploring machine learning algorithms to analyze data from cows with disorders monitored by an automatic monitor system and milking system across their entire lactation in intensive Chinese dairy farms.

The sensitivity of each model was not exceeded by 85%, with 48.15%, the lowest of the kknn algorithm. Although the specificity reached 77.78%, we considered this metric as poor. The performance of Rpart still needs improvement, especially for the sensitivity, which the farm staff are more interested in when the predicting model is applied to practice. In fact, we also conducted artificial neural networks and Bayesian networks to construct a predictive model, while none of the six measuring criteria exceeded 0.4. With the sample increased and the algorithm optimized, the model would perform better when considering the specific period (e.g., the transition period, the first from 1 to 3 weeks after calving, etc.) and blood or urine samples of subgroups of cows involved. In future work, we will intend to expand the data dimension to include e.g., biochemical, physiological, environmental, and farm management, and the algorithms will improve over time using artificial intelligence. Furthermore, we will attempt to apply the validated technology to a number of farms with different geographical and environmental settings. An improvement in predicting health disorders in dairy cows would benefit both humans and animals.

## 5. Conclusions

We used eight machine learning algorithms for analyzing a dataset of 14 dimensions regularly produced by the automatic monitoring system and milking system, and milking systems in intensive Chinese dairy farms. We presented six parameters to assess the performance metrics of the models, with the Rpart algorithm outperforming others and indicating its strong generalization ability. Future research should focus on applying artificial intelligence to transfer the algorithms to management software for accurately and precisely predicting and detecting disorders in lactating cows and calves. This could be achieved through the mining of greater reliable features in animals with diseases.

## Figures and Tables

**Figure 1 animals-12-01251-f001:**
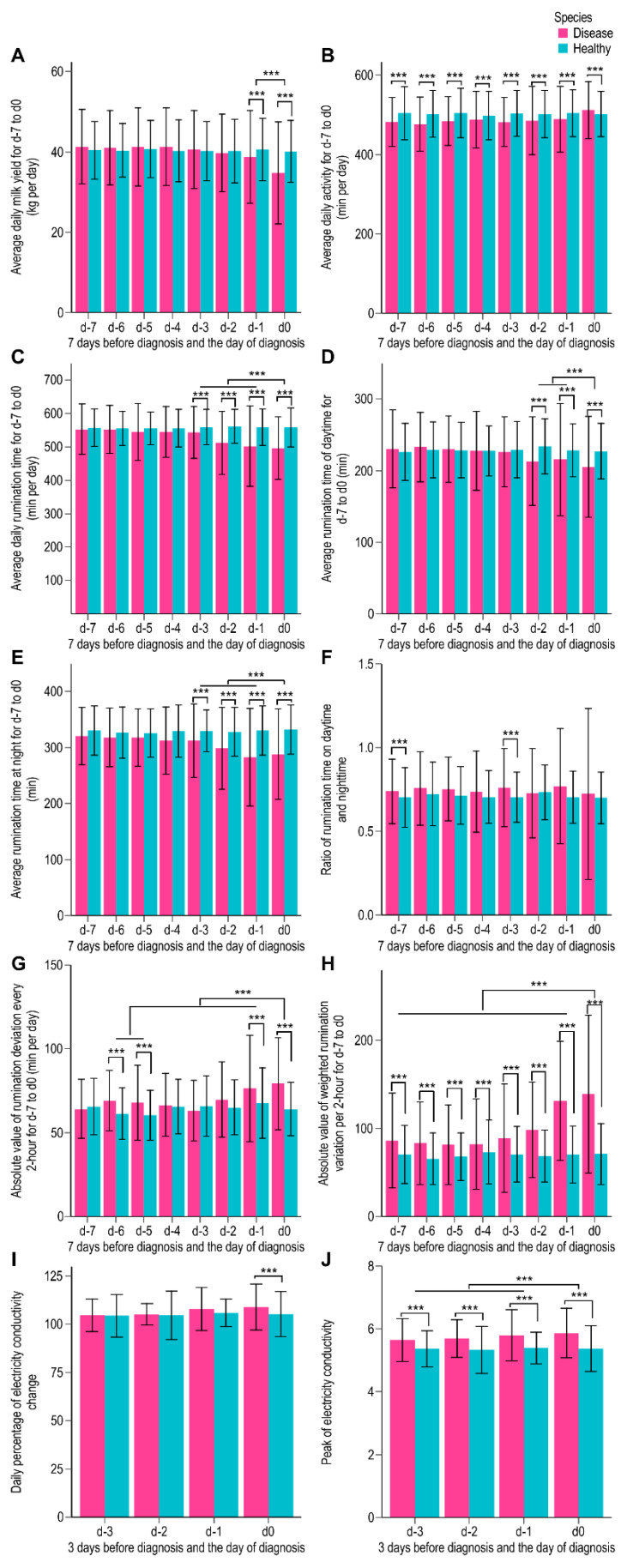
Box plots with the error bar and significance of each variable for cows with health disorders and healthy ones. Subgraphs (**A**) depicts the difference of daily milk yield of the cows with disorders and the healthy ones, (**B**) the difference of daily activity of the two groups, (**C**) the difference of daily rumination time of the two groups, (**D**) the difference of rumination time at daytime of the two groups, (**E**) the difference of rumination time at nighttime of the two groups, (**F**) the difference of the ratio of the rumination time at daytime to nighttime of the two groups, (**G**) the difference of the absolute value of rumination deviation every 2 h of the two groups, (**H**) the sum of the absolute value of the weighted rumination variation of the two groups, (**I**) the daily percentage of the change in the electrical conductivity of milk of the two groups, and (**J**) the peak electrical conductivity of milk of the two groups, respectively, at a significant level of 0.001. The *x*-axis is defined as the time from d-7 to d0, i.e., 7 days before the diagnosis is denoted as d-7, 6 days before diagnosis denoted as d-6, 5 days before the diagnosis is denoted as d-5, 4 days before the diagnosis is denoted as d-4, 3 days before the diagnosis is denoted as d-3, 2 days before the diagnosis is denoted as d-2, 1 day before the diagnosis is denoted as d-1, and the diagnosis day is denoted as d0. The *y*-axis presents the values of these variables. Violet-red and blue represent the “disordered” and “healthy” groups, respectively. The error bars represented standard deviations of the value of each variable on d-7 or d-3 to d0. “***” represented the difference of the two groups was at the 0.001 level of significance.

**Figure 2 animals-12-01251-f002:**
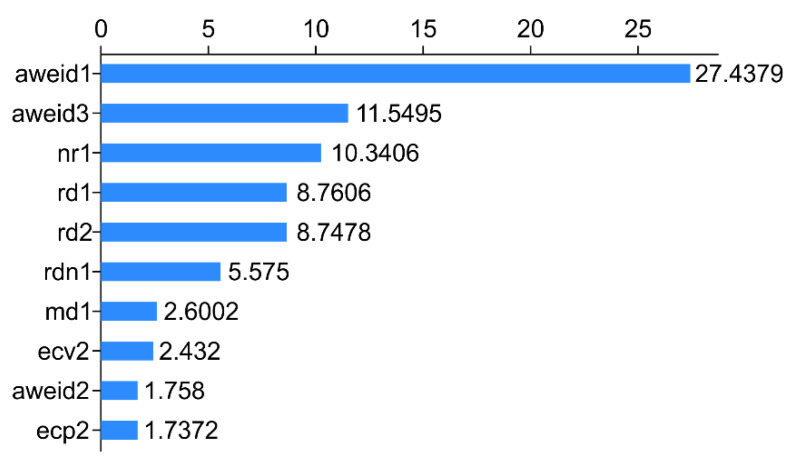
Importance of features for the ten variables in decision tree classification.

**Figure 3 animals-12-01251-f003:**
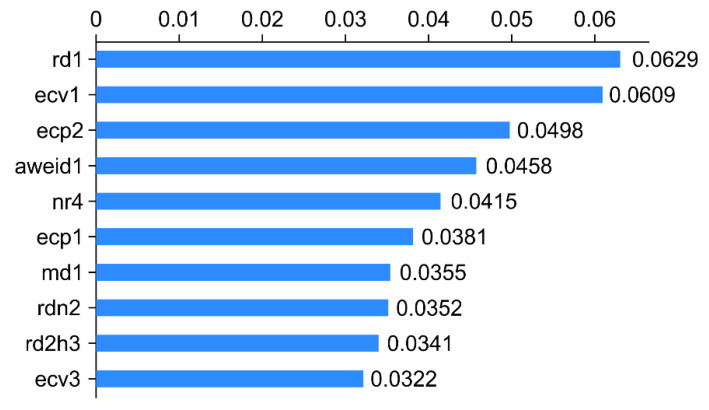
Importance of features for the ten variables in eXtreme Gradient classification.

**Figure 4 animals-12-01251-f004:**
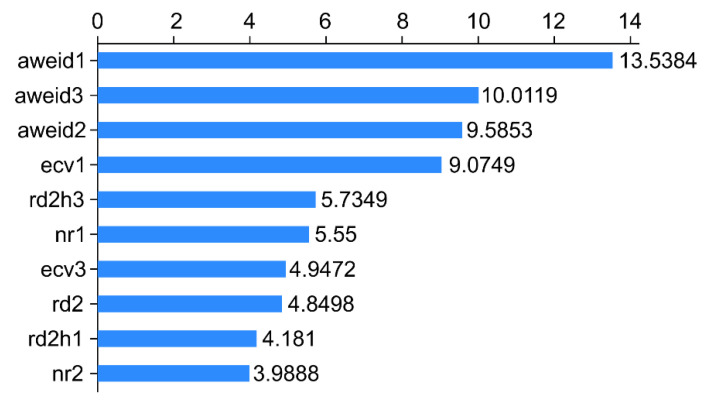
Importance of the features for the ten variables in Adaboost classification.

**Figure 5 animals-12-01251-f005:**
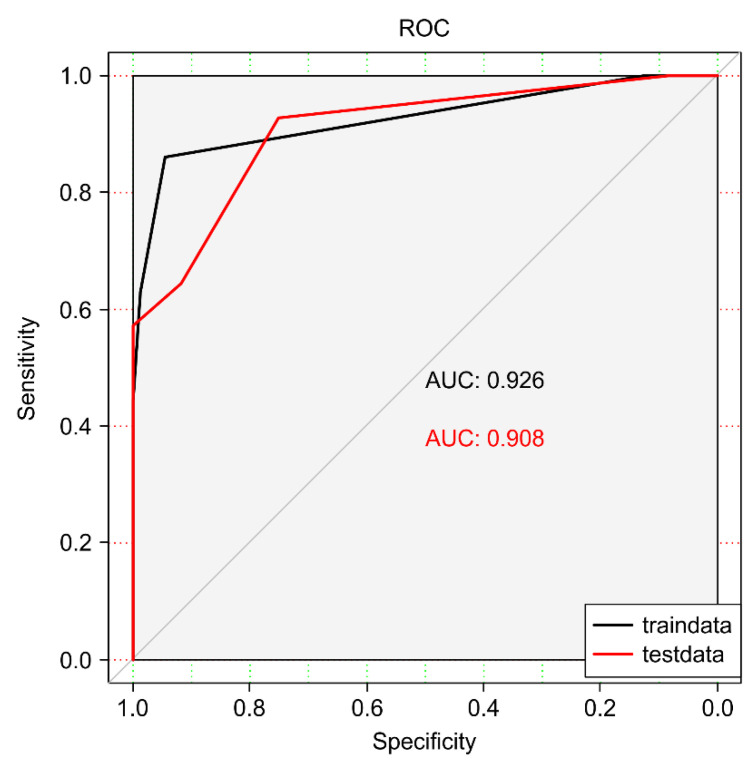
Receiver operating curve (ROC) of train data (75%) and test data (25%) of the Rpart algorithm.

**Table 1 animals-12-01251-t001:** Sources of information (automated monitoring system and milking system) and traits measured with variables at different measurement intervals, summed up to the daily values.

Sensor	Trait	Variable	Measurements Interval	Unit
Automatic monitoring system	Activity	Activity	min/2 h	min/day
	Rumination	Daily Rumination time	min/2 h	min/day
		Rumination deviation per 2-h	min/2 h	min/day
		The sum of absolute values of the weighted rumination variation	No./2 h	min/day
		Rumination at daytime	min/2 h	min/day
		Rumination at nighttime	min/2 h	min/day
		The ratio of rumination time at daytime to that at nighttime	No./2 h	Non
Milking system	Milk yield	Daily milk yield	kg/day	kg/day
	Electrical conductivity of milk	Daily percentage of change of the electrical conductivity of milk	No./milking shift	Non
		peak electrical conductivity of milk	mS/cm/milking shift	mS/cm

**Table 2 animals-12-01251-t002:** Performance of eight machine learning algorithms with six measuring criteria.

Model Performance	Sensitivity	Specificity	Accuracy	Precision	F1-Score	AUC (Confidence Interval)
Logistic	0.6071	0.7143	0.6667	0.6296	0.6182	0.685 ([0.576, 0.794])
SVM	0.7857	0.8750	0.8421	0.7857	0.7857	0.744 ([0.598, 0.890])
Rpart	0.6842	0.9474	0.8158	0.9286	0.7879	0.908 ([0.723, 0.930])
Random forest	0.8333	0.8462	0.8421	0.7143	0.7692	0.854 ([0.695, 0.951])
eXtreme Gradient	0.5882	0.8056	0.7358	0.5882	0.5882	0.828 ([0.714, 0.942])
Adaboost	0.8000	0.7857	0.7895	0.5714	0.6667	0.744 ([0.598, 0.890])
Naïve Bayes	0.8462	0.6800	0.7143	0.4074	0.5500	0.676 ([0.574, 0.778])
kknn	0.4815	0.7778	0.6508	0.6190	0.5417	0.630 ([0.511, 0.748])

AUC, area under the receiver operating characteristic curve.

## Data Availability

Data will be made available upon request to the corresponding author.

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
