# Peer review of "The Early Prediction of Common Disorders in Dairy Cows Monitored by Automatic Systems with Machine Learning Algorithms"

_animals, 2022, doi:10.3390/ani12101251_

Round 1
Reviewer 1 Report
Comments to author
Overall
Interesting study that using common on-farm measures to detect health disorder in lactating dairy cows. By grouping udder, reproductive and hoof disorders the authors were able to formulate machine learning algorithms to detect deviations from behaviour when a cow transition from healthy to sick, taking both daily duration and rhythmicity into account. The manuscript is well written and easy to follow. The manuscript would merit from a few clarifications, detailed below.
M&M
Please clarify the type (acceleration data?) of measures in Line 123-127: Data of the physical activity and rumination time recorded with the HR-Tag monitoring system (SCR Engineers Ltd., Netanya, Israel) were averaged and stored at 2-h intervals up to 24 h. We recorded the activity as a unitless measure of upward vertical head and neck movements, such as walking and mounting, while excluding downward vertical and horizontal movements, such as feeding.
Please clarify line 148-151: not the herd average but an average of healthy cows in similar stage of lactation and parity was used for controlling for the environment in the algorithms?
Please state how many observes (staff and vet) and inter-observer agreement if available, if not please comment (line 154- ).
How long time from detection (farm staff) to diagnosis (vet)? Line 154-.
Please clarify line 157 (every milking, daily?): Clinical signs of mastitis were regularly examined by observing the udder
Please clarify line 165: We did not consider a cow sick more than once during lactation. Was she removed from the trial after one sickness event?
Line 224-227: I am not about the use of control cows, can you please clarify this? I would assume that the cows are their own controls as you are looking for the deviation in behaviour when a cow transition from healthy to sick.
Results
Line 247-248: Am I right to interpret the results as that you simply used the control cows as comparison, instead of including a herd average in the algorithm to control for external factors such as hoof trimming or other disturbances? Could you please comment on this approach, thank you.
Figure 2-4: please add explanation to variables and x-axis in the figure caption. It would be beneficial if the abbreviation is written in direct connection with the explanation.
Reviewer 2 Report
The authors present a high quality paper where they predict the risk of disorders in dairy cows considering 14 variables and comparing eight machine learning algorithms. The methods and analysis are advanced, and the paper is generally well written. I really enjoyed reading it and I only have a few minor suggestions to improve the paper.
Line 64-73. Here you make a list of studies that used machine learning algorithms but you do not really provide why they used different algorithms, and what are the advantages of them. It would be interesting for a general audience to have more background on these algorithms.
Line 80. You should clarify why you used these eight algorithms. At this point of the paper a reader does not have enough background to understand why you plan to compare eight algorithms, and what are the advantages/disadvantages of each of them.
In results and figures, you should specify what the variability and error bars show (I assume standard deviations but that should be specified)
Reviewer 3 Report
Regardless of the formulation of the general purpose of the research / study in the Introduction chapter, the article should also take into account what was the cognitive (scientific) goal of the research, and what utilitarian (useful) goals were formulated by the authors in the research? The overview of the knowledge in the Introduction chapter could be summarized in the form of a sentence describing the research problem. One can write: "The research problem is ...". At the same time, the research problem could be associated with an indication of a gap in the current state of knowledge, which was an inspiration to conduct research and obtain the results presented in the article.
Perhaps it would be worth writing what the errors (error values) were in the analysis performed. Did the sizes of these errors differ between the tested models / algorithms?
I have doubts about the sentence on lines 104-105. The authors reported that the cows were milked three times a day, including the milking hours. At the same time, in the same sentence, the authors wrote that an automatic milking system was used. In my opinion, there is an inconsistency regarding the information provided here. The wording "automatic milking system" refers to the voluntary milking of cows, so giving specific milking times is misleading. I think the authors were talking about a solution where the cows are milked three times a day in the milking parlour, and the equipment in the milking parlour includes devices for automatic cluster removal, automatic dipping and more. Therefore, in my opinion, the sentence in lines 104-105 should be rewrited so as not to arouse controversy among the reader. Regardless of this, I am not sure if the term "automatic milking system" used by the authors in the text is correct in relation to the actual solutions used in the researched dairy farms. Were the cows really milked with AMS?
In the text of the article, it would be worth writing whether the animals included in the experiment were kept in accordance with the principles of welfare. In other words, whether the welfare of the animals was maintained in the place where they were kept. In general, the authors wrote about the conditions in the barn, but it would be worth summarizing it by saying whether the animals were provided with an appropriate level of welfare.
In the description of the research material, it would be good to add information on the average age of the cows included in the research.
How was the poor appetite of cows (line: 163) tested in the experiment? Have special (research) feeding stations been used for this purpose to monitor the amount of feed consumed by the cows?
In Table 1, it would be nice to organize some information in the lower part. In the "Variable" column, at the very bottom, you probably need to separate two sentences if they relate to two records in the next "Measurements interval" column. The first of these sentences begins with "Daily percentage ...". If the word "percentage" is included, why is there no "%" in subsequent columns? As for me, this part of the information at the bottom of Table 1 creates a huge information confusion and it would be worth putting it in order.
I would like to ask, for which herd sizes of dairy cows are the analyzed animal health hazard identification systems applicable? Is there any limit to the size of the herd of cows? Perhaps it would be worthwhile to write in the discussion of the results of the research, what are the limitations in the implementation of the presented idea of ​​studying a herd of cows, taking into account the number of animals and access to specialized programs and equipment of farms with devices monitoring changes in indicators characterizing animals, their production indicators and health.
Reviewer 4 Report
Manuscript animals-1687881, entitled “The Early Prediction of Common Disorders in Dairy Cows Monitored by Automatic Systems with Machine Learning Algorithms”
Recommendation: The above paper is not suitable for publication in its present form.
The article provides useful information about the early prediction of common disorders by monitoring automatic systems with machine learning algorithms in dairy cows. Although, the experiment was in general appropriately designed and implemented, there are some points that should be corrected or clarified.
General comments
- In Material and Methods, please provide the definitions for “rumination deviation per 2 h (unitless)” and “the sum of the absolute values of weighted rumination time variation”.
- L104: What do you mean by automatic? Robotic? In robotic systems cow are milked whenever they want. In conventinal milking systems, milker put the cups, so animals are not automatically milked.
- In Figures 2-4, please explain abbreviations as a footnote. At the same time, please add Figure in a parenthesis at the end of each paragraph (L308-316, 320-327, 331-339).
- At the end of the discussion, please add a paragraph with the limitations of your study.
Minor points
L18: “…the whole lactation period in…”
L30: “days in milking”
L54: “linked” instead of “related”
L60-61: “…the negative effects on health and performance of dairy cattle. Thus…”
L62: “…than clinical signs…”
L77: “…bovine diseases, and particularly an early warning system using the data…”
L103: What do you mean by “with the feed provided whenever necessary”? Two meals or whenever necessary?
L138: What do you mean by “for reproductive events and survival”?
L148: “…of milk in healthy…”
L227: Please provide the respective values
L241: “…was higher than..” When? d-0?
L247-248: What do you mean? Healthy control cows? Please clarify
L271-272: Estrus display was not affected by health disorders?
L367: Please delete “on”
L371: What do you mean by “test day”?
L387: “…higher proportion of detection in early postpartum cows. The first…”
L390: Please delete “respectively”
L395: To evaluate metritis treatment, de Oliveira…”
L397: “included to” instead of “fed into”
L402: “The KNN performed the best…”
